# Understanding the Dynamics of Inflammatory Cytokines in Endodontic Diagnosis: A Systematic Review

**DOI:** 10.3390/diagnostics14111099

**Published:** 2024-05-25

**Authors:** Ignacio Barbero-Navarro, Maria Esther Irigoyen-Camacho, Marco Antonio Zepeda-Zepeda, David Ribas-Perez, Antonio Castaño-Seiquer, Iuliana Sofian-Pauliuc

**Affiliations:** 1Dental School, University of Seville, 41009 Seville, Spain; ibarbero2@us.es (I.B.-N.);; 2Health Care Department, Autonomous Metropolitan University-Xochimilco, Mexico City 04960, Mexico

**Keywords:** root canal therapy, root canal debridement, root canal treatment, endodontics, cytokines

## Abstract

The primary aim of this literature review is to delineate the key inflammatory cytokines involved in the pathophysiology of pulp inflammation. By elucidating the roles of these cytokines, a deeper comprehension of the distinct stages of inflamed pulp can be attained, thereby facilitating more accurate diagnostic strategies in endodontics. The PRISMA statement and Cochrane handbook were used for the search strategy. The keywords were created based on the review question using the PICO framework. The relevant studies were meticulously assessed according to predefined inclusion and exclusion criteria for this systematic review. A rigorous quality checklist was implemented to evaluate each included study, ensuring scrutiny for both quality and risk-of-bias assessments. The initial pilot search conducted on PubMed, Scopus, Cochrane, and WoS databases yielded 9 pertinent articles. Within these articles, multiple cytokines were identified and discussed as potential candidates for use in endodontic diagnosis, notably including IL-8, IL-6, TNF-α, and IL-2. These cytokines have been highlighted due to their significant roles in the inflammatory processes associated with pulp pathology. The identification of specific inflammatory cytokines holds promise for enhancing endodontic diagnostic procedures and exploring diverse treatment modalities. However, the current body of research in this area remains limited. Further comprehensive studies are warranted to fully elucidate the potential of cytokines in refining diagnostic techniques in endodontics.

## 1. Introduction

The tooth is composed of rigid mineralised materials (enamel, cementum, and dentine with fluid-filled tubules) and soft tissue called the pulp. This tissue contains blood vessels, nerve bundles coming from the apical region (the end of the root), odontoblasts, mesenchymal progenitor cells, and immune cells [1].

The pulp and its components keep the tooth mineralised, nurtured, immunocompetent, and innervated (it is linked with the central nervous system, giving a sensory output). This gives better resistance and functionality to the tooth for longer survival [2].

The tooth can self-repair and block those infected tubules. This forms a “reactionary” barrier between the microorganisms and the pulp when caries, trauma or cracks are present. This is possible because of the odontoblasts, which are believed to be the first cells to encounter the bacteria (they are connected to the dentinal tubules and pulp). They are capable of releasing signalling factors such as chemokines and cytokines which aim to kill the invaders. The pulp can heal if the triggers are removed in time and the contamination is minimal. The term to describe this would be reversible pulpitis, as there would not be signs or symptoms to indicate that the pulp is incapable of healing [3,4,5,6].

However, this is a limited response due to the small and limited space a tooth has in the oral cavity. If the triggers persist and reach the pulp tissue, which is very vulnerable, the inflammatory immune response will damage it irreversibly. This will lead to pulpal pain, irreversible pulpitis, pulpal necrosis, or affect the periapical tissues and evolve into apical periodontitis [AP] [7,8,9,10,11].

### 1.1. Inflammation

Inflammation is a vascular response alongside changes in the mediators’ cells. The change from reversible to irreversible pulpitis depends on the accumulation of immune cells and how long is prolonged in time.

Once the pulp has died, the periapical tissues will become inflamed and become chronic, producing bone resorption, abscess, or granuloma [12,13]. The level of pulp inflammation is related to the size and type of exposure along with the duration and severity of symptomatology in the patient [9].

This type of inflammation is stimulated by the lipopolysaccharides (LPS) or endotoxins that come from the bacterial walls and induce the activation of immune cells. They secrete pro and anti-inflammatory cytokines and other tissue-degrading enzymes [3,14,15].

In the preliminary stages, the inflammatory mediators, such as cytokines, can induce repair as mentioned before. The tissue environment is disrupted if the expression of the mediators and inflammation are sustained in time, leading to the deactivation of repair. This can create pulp necrosis and periapical inflammation that can lead to bone resorption or AP [16,17,18].

### 1.2. Immune System

The inflammatory cells form part of both the innate and adaptive immune systems. This is important because depending on the type of response that comes from the immune system, the injured tissue will heal or will be more damaged [16].

When the inflammation is mild or in its earlier stages, it is the innate immune system that acts as the first line of defence [19]. As stated before, if this inflammation becomes chronic and the infectious agent is not removed, a more specific response is needed by shifting from the immune system to adaptive immunity [6,15].

The innate system mainly comprises polymorphonuclear neutrophils (PMNs) and monocytes that activate a cascade of proinflammatory cytokines [12,18]. The adaptive immune system provides a specific response to the infection in the tooth, which aims to avoid dissemination beyond the periapical tissues to the rest of the body. The most important cells that form part of this system are T and B cells [5,20].

The final stage of the adaptive immune system response is when the infection in the dental tissue has been removed [6]. On the other hand, if the host defence mechanisms fail to eradicate the infection, chronic inflammation and bone resorption will take place and persist [11].

### 1.3. Cytokines

Cytokines are small signalling molecules [glycoproteins (10–15 kD) [6] that mediate and regulate host responses (intensity and duration) to infection, inflammation, and trauma. They maintain the equilibrium to avoid persistent inflammation. Proinflammatory cytokines initiate or enhance systemic inflammation while anti-inflammatory cytokines reduce inflammation and promote healing [12,18]. Their course of action (reparation or destruction) depends on their context and concentration [21].

These cytokines are produced by immune and non-immune cells, and they can induce many interactions, such as proliferation, differentiation, or apoptosis. This group consists of interleukins (IL-), interferons (INF-), lymphokines, tumour necrosis factors (TNF-), and chemokines [6].

Examples of proinflammatory cytokines include IL-1 and TNF-α. For the anti-inflammatory type, IL-10 is a good example. Some cytokines, such as IL-6, are multifunctional and can have both proinflammatory and anti-inflammatory properties [12,21]. Lipopolysaccharides are the most potent stimuli for immune cells regarding the release of several inflammatory mediators (e.g., IL-1α, IL-1β, TNF-α, IL-6, PGE2, IL-10, and MMPs) [10,20].

The goal is to create an environment absent of inflammation and low microbial load so the periapical tissue can heal, repair, and regenerate [16]. There is a consensus in trying to move from root canal therapy to preserving the pulp as much as possible or regenerating the pulp tissue [5,14].

### 1.4. Diagnosis

For an accurate endodontic diagnosis, the clinician needs knowledge about pulp inflammation and how this would change depending on the endodontic procedure applied. Unfortunately, the tools to assess pulp inflammation are limited. A better assessment would be delivered to the patient if molecular strategies were introduced [8,10].

The difference between healthy and necrotic pulps has been significantly studied but without focusing on specific conditions of the inflamed pulp and its viability for vital pulp therapy (VPT). It is often the case that, in clinical settings, patients undergo restorative treatment with an uncertain prognosis of the pulp’s vitality, if there are no signs or symptoms. The opposite happens when pain and other symptoms appear when endodontic treatment is offered, often leading to overtreatment when a minimally invasive treatment could have been performed. It is still difficult to change clinical judgment to encourage preserving the pulp’s vitality and natural structure of the tooth. It is because of this, that research in studies that assess the health of the dental pulp through inflammatory mediators is important and could give a potential clear guidance [22].

It is currently advised that “irreversible pulpitis” does not have to mean necrosis of the pulp because it does not always match the pulp’s biology or its response. There is still general doubt on classifying pulpitis (irreversible, partial irreversible, reversible) when the histological findings do not meet the symptoms or signs that the patient presents with [23,24].

In fact, Hahn and Liewehr in 2007 [25] describe how the intensity or appearance of the pain depends on the type (Gram-positive or Gram-negative) and quantity of bacteria present in deep carious lesions. They explain that based on this, different proinflammatory cytokines will be synthesised and, therefore, some inflammatory mediators induced by these cytokines sensitise C fibres in the dental pulp, explaining the prolonged pain to heat testing in deep caries. On the other hand, when there is no thermal sensitivity in the same scenario (deep caries and vital pulps), they induct anti-inflammatory cytokines, such as IL-10, and the suppressing effect of the organic acids directly produced by *Lactobacilli* bacteria.

The pulpal inflammatory response will also depend on the remaining dentine thickness and dentinal tubular permeability. Its thickness can help reduce the concentration of bacteria reaching the pulp and, thus, induce repair and health instead of more inflammation and necrosis. If the thickness is above 2 mm from the pulp, they can recover amazingly fast and, if below 1.5 mm, the inflammatory cells would increase [25]. In clinical settings, this dentinal thickness is usually measured through a radiograph, despite their underestimation of the depth of carious lesions as this is the quickest method to assess the pulp’s proximity to the lesion [26].

This review of the literature will analyse the cytokines profile (type and quantity) during the distinct stages of the pulp inflammation alongside its severity (how soon it starts and how much of the pulp is inflamed). With a clearer view, a diagnosis would be easier and better for the clinician as they would have a guideline and, in certain scenarios, preserving the vitality of the tooth would become the first line of treatment. Considering the diagnosis is accurate, inflammation would be resolved in its initial stages and not left to persist and become chronic.

The objective of this literature review is to enhance comprehension regarding the response of dental pulp to external aggressions, elucidating its pathophysiological manifestations and delineating the intricate involvement of diverse cytokines. Through the identification of these cytokines, this review aims to facilitate an enhanced diagnostic approach, thereby enabling the future implementation of minimally invasive endodontic procedures aimed at preserving the vitality of the tooth.

## 2. Materials and Methods

The literature review carried out during this stage aimed to find a gap in diagnosis in endodontics and if a literature review was suitable. This preliminary research led to a “descriptive” research question for this intended study: the influence of inflammatory cytokines in the endodontic diagnosis.

### 2.1. Criteria for Considering the Selected Studies

A specific and focused topic for a literature review comes from a well-formulated research question with a clear definition of the eligibility criteria [27].

A PICO (Population, Intervention, Comparison, Outcomes) tool was used to identify the components of the review question. The following conditions were applied:-Patient or Population: children with permanent dentition and adults of all ages presenting reversible or irreversible pulpitis, symptomatic apical periodontitis, asymptomatic apical periodontitis, and/or reinfection of a previous root canal treatment.-Type of intervention: diagnosis based on the identification of cytokines (type and quantity), pathology of the pulp and symptomatology in a review on dental examination, emergency, and additional tests such as radiographs.-Comparison: how studies analyse inflammation, cytokines, and pathology to reach a diagnosis: systematic reviews, meta-analysis, randomised controlled studies, and controlled clinical trials. These analyse the extirpated pulp by extracting it partially and treating the rest or by extracting the tooth.-Outcome measures: modification of the endodontic diagnosis in daily practice. Different approach to investigate an inflamed pulp.

### 2.2. Search Strategy and Databases

The PRISMA (Preferred Reporting Items for Systematic Reviews and Meta-Analysis) flow chart was followed and customised [28]. Terms used:PubMed: “root canal therapy” or “root canal debridement” or “root canal treatment” or “endodon*” AND “inflamm*” AND “cytokin*”.Other combinations in other databases: “root canal therapy” AND “inflammation” OR “cytokine” AND “endodontic*”.(“pulp inflammation” OR “dental pulp”) AND (“inflam*” AND “cytokin*”) AND “diagnosis”.(“pulp NEXT inflammation” OR “dental pulp”) AND (“inflam*” AND “cytokin*”) AND “TREATMENT”

### 2.3. Study Selection

Two independent reviewers (I.S.) and (I.B.) scanned twice to ensure there was not a missed study. When any ambiguity was found, it was solved by asking a supervisor for their advice (D.R.). Duplicated studies will be removed first and recorded.

The remaining studies were scanned in more depth by reading the full text and assessed against the pre-set inclusion and exclusion criteria (second stage searching) [29]. The Preferred Reporting Items for Systematic Reviews and Meta-Analysis (PRISMA) study flow diagram is used to show the search process and results [30].

#### 2.3.1. Included Studies

-Articles that are Level A or B of evidence-based systematic reviews, meta-analysis, randomised controlled trials, controlled clinical trials, and cohort studies. Articles published in peer-reviewed scientific journals in the English language, published between 2014 and 2023.-Articles will be related and relevant to the title of the systematic review: “Influence of inflammatory cytokines in the endodontic diagnosis”.-These articles need to be related to inflammation (including pulpitis and apical periodontitis).-Articles found by citation chaining and relevant to the topic will be considered.

#### 2.3.2. Excluded Studies

-Studies that do not meet the inclusion criteria. Low-level evidence studies (Levels C, D, E): case reports, case series, case–control studies, cross-sectional studies, and expert opinions.-Articles related to systemic inflammation and diseases (non-endodontic related such as Arthritis).-Articles related to inflammation and periodontitis (commonly known as “gum disease”).

### 2.4. Data Extraction and Management

A data extraction table was created using predefined data fields based on the predefined inclusion criterion.

One author extracted the data (I.S.), and in case of uncertainty, a supervisor was consulted (D.R). The results were summarised and tabulated clearly with RevMan software 5.4.

### 2.5. Quality Assessment

-Two reviewers made this stage following the JBI (The Joanna Briggs Institute) appraisal tool. It evaluates the quality of several types of studies, such as cohort studies or systematic reviews [31].

## 3. Results

The initial pilot search with the PICO tool was conducted in all the databases. This search initially resulted in 239 articles before applying the filters mentioned earlier. Once these were added, 195 articles were found to be relevant for screening the titles and abstracts by applying the review question and the inclusion criteria. After the screening, 33 articles were selected in the main search for further analysis, ending with 9 relevant articles that were tested for the eligibility and inclusion stage (Figure 1).

The main results are summarised in Table 1.

Null hypothesis: According to the authors, the success of vital pulp therapy treatment based on the diagnosis will not be influenced by age, nor will there be a limit as to what treatment is best indicated for the pulp in specific scenarios. However, this could be different because some patients have systemic diseases or are under medication, which could influence the diagnosis and treatment outcome or limit the immune system response.

There is an insight in the articles selected into the most important cytokines during the pulp’s inflammatory response, its assessment, and its understanding of the inflammatory mechanisms. Cytokines are considered as possible biomarkers of the pulp’s inflammatory status as a diagnostic tool for predicting and differentiating the pulp’s healthy or disease state in order to prevent over-treatment. The studies selected were considered because of the presentation about the cytokines in pulpal inflammation, its high level of evidence, and clarity on the information presented. This will be displayed below along with a table to differentiate the different cytokines involved depending on the pulp’s health state.

The proinflammatory cytokines mentioned to be useful as a diagnostic tool in pulpitis and acute inflammation were IL-1 β, TNF-α, IL-6, and IL-8.

Overall, there are two groups of cytokines:

The first type (proinflammatory) includes IL-2, IL-8, TNF-α, and Interferon-y. These appear in the pulp’s proinflammatory process and stimulate a strong cellular immune response, particularly the phagocytic type. Their quantity and quality will influence the evolution of inflammation.

The cytokine IL-8 creates neutrophil chemotaxis and releases degradation enzymes. It is thought to be a primary regulatory cell in the acute phase of the pulp’s inflammation. Its presence will only exacerbate the inflammatory process.

TNF-α stimulates acute inflammatory proteins, induces the production of inflammatory cytokines, and increases leucocyte toxicity. However, this cell is pleiotropic, and it is also essential in the anti-inflammatory process when the pulp responds to infection. This cell will act as a pro- or anti-inflammatory depending on its concentration or time context.

Unfortunately, there is no knowledge of the cut-off point for when the TNF-α expression will inhibit the pulp’s healing or start its repair. TNF-α along with IL-1, can also play a key role in the development of apical periodontitis and periapical tissues. As a result, these could be useful biomarkers for determining the extent of inflammation.

The second type would be those cytokines that induce repair or healing of the pulp, such as IL-10 or IL-4. These induce B-cell proliferation and differentiation into plasma cells, suppressing further phagocytosis activation.

The application of biomaterials such as hydraulic calcium silicate types of cement (HCSCs), including Biodentine or MTA, in pulpotomies helps induce tissue repair through their properties. It is thought that these materials lower proinflammatory cells such as IL-1 α, IL-2, or IL-6, indicating their anti-inflammatory properties. These cements also have antimicrobial properties in their initial settings due to their high pH (10.2–12.5).

The articles selected desire the implementation of quantifying the cytokines levels during acute pulp inflammation and root canal treatment, or after finalising treatment.

There is also a description and analysis of cytokines, T cells, macrophages, RANK-L, the OPG system, and other inflammatory markers to give a comprehensive understanding of the creation of apical periodontitis.

The cells more predominant in apical periodontitis are IL-1 α, TNF-α, or IL-6, which activate the osteoclasts (involved in the bone resorption or destruction) and fibroblasts. IL-17 is mentioned due to the possible attraction of neutrophils and induction of the RANK-L system by activating osteoclasts. Although its role is not fully understood yet.

IL-6 is also an important cytokine because, if there are elevated levels of this cell, the periapical lesions seem to be symptomatic and active. TNF-α seems to be related to the quantity of macrophages in the surrounding tissues of the apical periodontitis, which creates more vascularisation and inflammation.

Finally, the RANKL/OPG system is a bone regulator that influences the differentiation and activity of osteoclasts. RANKL works similarly to TNF-α and osteoprotegerin (OPG) can block or prevent osteoclastic differentiation or its activation, reducing bone resorption. This system is important because it creates an inflammatory process in the periapical tissues after the pulp has necrosed, giving the clinician a better understanding of the formation of chronic inflammatory processes and apical periodontitis (Table 2).

## 4. Discussion

Clinically, a healthy pulp can be defined as a pulp without any signs or symptoms of disease. The level of pulp inflammation is often compared to the severity and duration of the stimulus and the host’s response. Because of the symptoms (thermal pain, spontaneous or referred pain), clinicians often diagnose an inflamed pulp based on subjective and objective findings without fully understanding how the inflammatory response of the pulp works.

This is also challenging because the diagnostic tests used in daily practice are limited to assessing the inflammatory response and state of the pulp and periapical tissues (pain quality, history and pulp sensitivity tests are often used to evaluate this). Molecular testing is promising in the endodontology field because it could clarify what type of cytokines and inflammatory mediators could be useful in order to differentiate the different stages of inflammation and, thus, set a foundation for testing and outcome measures for a healthy or diseased pulp [32,33,34].

Recent findings using molecular techniques, such as high-throughput transcriptional profiling of carious pulpal tissue, suggest that the majority of diseased teeth are caused by proinflammatory processes rather than molecular events linked to healing [32].

Subsequent analysis of these data revealed the presence of a chemical termed adrenomedullin, which is unique to pulp biology and is upregulated during dental disease. This molecule is classified as a pleiotropic growth factor/cytokine that may induce mineralised tissue differentiation, secretion, and angiogenic processes. It also possesses antimicrobial and immunomodulatory qualities [32,33,34].

There are several techniques for evaluating these highly elevated proteins’ expressions in pulpal blood. The multiplex test (using LUMINEX), radioimmunoassay, enzyme-linked immunosorbent assay, Western blot, and fluorometric assay are among the techniques. Because of its multiplexing capabilities, versatility, high precision and accuracy, customising choices, and efficiency in terms of both time and sample volume, the Luminex instrument is preferred when used in conjunction with multiplex assays [34].

With a more accurate diagnosis, the pulp could maintain the vitality of the tooth more often than nowadays. This will only occur if the clinician has knowledge about the stages of pulp inflammation, the successful elimination of bacteria and the response to endodontic procedures [3,7,33,34,35].

Improper diagnosis of the pulp can lead to clinical failures when trying to implement minimally invasive endodontic procedures such as indirect pulp capping, stepwise caries removal, direct pulp capping, or pulpotomies. The failure of these procedures is multifactorial, often related to poor pulp evaluation or underestimating the severity of pulp inflammation.

Despite that, The American Association of Endodontists (AAE) has created a criterion where the clinician can classify pulpal disease; this is limited as histologic evaluation can show no correlation between clinical diagnosis and in situ pulp status. This shows how unreliable a diagnosis and the unpredictability of treatment outcomes on vital pulp therapies can be, making the practice of root canal treatment more frequent than is necessary due to its success and predictability [10,32,34].

The real question would be why Philip and Suneja in 2022 [24] and Kahler et al. in 2023 [23], in addition to the endodontic associations, did not include the importance of the cytokines and other biomarkers when describing the histopathology assessment, let alone the inflammatory process at a more microscopic view in the diagnosis and treatment of the pulp.

The desire to maintain a vital tooth has prompted looking for new diagnostic tools and methods, such as measurement and identification of cytokines during pulp inflammation. This would enable the tooth to maintain its resistance to structural failures (fractures, losing sensory function, or biological defence mechanisms) and avoid progression of the inflammation and infection on the periapical tissues [11,12,13,14,33,34,35].

Numerous immune and non-immune cells produce and secrete cytokines, which are small glycoproteins (10–15 kD) that influence a variety of interactions between these cells, such as proliferation, differentiation, cell requirement, apoptosis, and many other actions. Cytokines are also known as interleukins, interferons, lymphokines, tumour necrosis factors, and chemokines [33].

Reaching a consensus on how to read and interpret the cytokine levels expressed per total amount of protein present in a sample (blood or tissue) and which types (the ones that appear most frequently are: IL-2, IL-6, IL-8, IL-10, and TNF-α, among others) is difficult when studies do not present standardised sample collection, measurements, study design, data analysis, or their quality. For this reason, a method that analyses the biomarkers in clinical settings has not yet been developed [19,21,32,33,34,35,36].

Further laboratory and preclinical investigations that support clinical trials are required to increase our understanding of pulpal inflammation and repair and to achieve an immunomodulatory strategy for therapeutic use. It is essential to have both appropriate animal models that support the proof of concept and two- and three-dimensional cell culture models that closely mimic the in vivo inflammatory state. Furthermore, it is necessary to create local tissue and topical drug delivery systems that enable the focused application and regulated release of these treatments in inflamed pulp regions. It is anticipated that local delivery of adjunct anti-cytokine treatment will be possible using appropriate carriers, such as hydrogel, dendrimers, and micro/nanoparticles [35].

One intriguing hypothesis is that in the future, diagnostics will make it possible to quickly ascertain the kind and intensity of inflammation within the pulp by utilising a panel of biomarkers that can detect and measure the presence of particular cytokines, as well as identify the type of bacterial flora and their byproducts [37].

With this information, treatment might then be customised utilising specific anti-cytokine medicines to the patient’s profile, leading to a more predictable therapeutic result. The clinical translation of this technology will, however, face several obstacles, such as the expense of development, the need for clinical trials, and the creation of suitable diagnostics. Indeed, distinct immunotherapies exhibit varying degrees of efficacy, as is seen in other chronic inflammatory illnesses with comparable clinical presentations.

Moreover, it may be necessary to determine the dosage required in various clinical scenarios. Considerable work therefore needs to be undertaken regarding this concept in order to determine the most appropriate cytokine targets for pulpal disease.

Indeed, to guarantee the maintenance of a host response against infection, it may not always be appropriate to target master inflammatory regulatory molecules. Instead, specific downstream targets may be more efficacious while maintaining adequate local host tissue protection. Notably, antibodies–cytokine fusion proteins, or immunocytokines, are now being developed and are emerging as novel therapeutics in the fields of cancer, autoimmune and chronic inflammatory diseases. These drugs have the potential to increase the therapeutic index, thereby increasing efficacy and outcome. It is possible to deliver adjunctive anti-cytokine therapy locally using a suitable carrier, such as a micro/nanoparticle, dendrimer, or hydrogel [36,38].

## 5. Conclusions

The continuous evolution in endodontics is not only changing the type of treatment but also the approach to the diagnosis. This review aims to show the clinician a more conservative approach in their daily practice when treating inflamed pulps by making an improved diagnosis and choosing an alternative to root canal therapy whenever possible (selective caries removal or vital pulp therapy are some examples of preserving the pulp vitality).

Inflammatory mediator expression levels are correlated with the development of clinically irreversible pulpitis. There are quantitative and qualitative differences in the way that reversible and irreversible diseases manifest. Our data suggest that in situ measurement of inflammatory mediators might help distinguish between permanent and reversible pulpitis clinically.

The possible cytokines to be used as biomarkers to assess the pulp’s state of health and disease would be IL-1 β, TNF-α, IL-6, and IL-8 as proinflammatory and IL-4 and IL-10 as anti-inflammatory cells.

While there have been advancements in understanding this topic, research remains constrained, highlighting the necessity for additional randomised controlled trials (RCTs) and longitudinal studies. Presently, the evidence is insufficient to select specific cytokines that offer precise diagnostic information regarding dental pulp status.

## Figures and Tables

**Figure 1 diagnostics-14-01099-f001:**
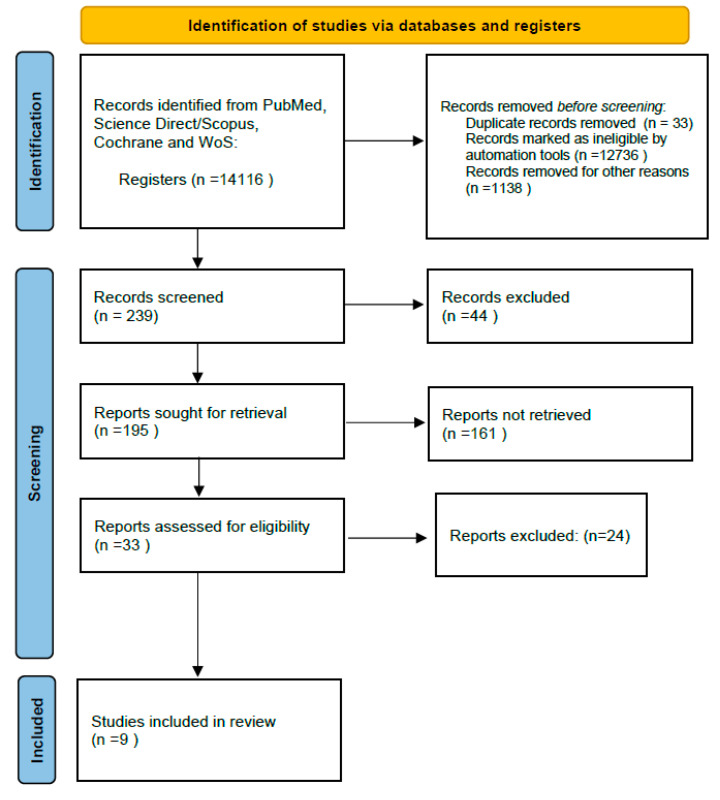
PRISMA flow chart of the study.

**Table 1 diagnostics-14-01099-t001:** Articles selected titles, authors, year of publication, objectives, and main results.

Article, Year, Author	Cytokines Studied	Objective	Results
The role of IL-6 on apical periodontitis: A systematic review. (2014) Azuma et al. [26].	IL-6	Examine the current knowledge of the role of IL-6 in apical periodontitis and if it could serve as a measure for differential diagnosis or as a biomarker that can further predict the progression of bone resorption.	IL-6 may increase levels of inflammation and reabsorbing bone in the presence of infection. IL-1 and TNF-α also induce bone resorptions. Further studies are needed to assess the relationship between specific cytokines and apical periodontitis.
Inflammatory profile of chronic apical periodontitis: a literature review. (2009, Braz Silva et al.) [12]	IL-17, TGF-β, IL-8, IL-6, TNF-α.	Review the inflammatory biomarkers related to apical periodontitis.	Different inflammatory cells and their byproducts are involved in the creation of apical periodontitis. The most important cytokines identified: IL-17, TGF-β, IL-8, IL-6, TNF-α.
Inflammation and regeneration in the dentin-pulp complex: A double-edged sword. (2014, Cooper et al.) [32]		Summarise and clarify the complex signalling during the invasion of bacteria, inflammatory cells, and immune system activation	The effects mediators are temporal context dependent. Further research is needed between inflammation and regenerative responses.
Role(s) of cytokines in pulpitis: Latest evidence and therapeutic approaches. (2017, Khorasani et al.) [33]	IL-8, IL-6, IL-1, IL-2.	Describe the role of cytokines in pulpitis.	Inflammatory cytokines play an important role and regulate the intensity of the immune response against infection. Some mentioned: IL-8, IL-6, IL-1, IL-2.
The Presence and involvement of interleukin-17 in apical periodontitis. (2019, Xiong et al.) [11].	IL 17	Reviews recent studies regarding the collective in vitro, in vivo, and clinical evidence of the presence and involvement of IL-17 in AP	Evidence for the presence of IL-17 in AP from human and animal models is clear. However, there is relatively little information currently available that would highlight the specific role of IL-17 in AP
Proinflammatory activity of primarily infected endodontic content against macrophages after different phases of the root canal therapy. (2015, Marinho et al.) [14]	IL-1β and TNF-α	Investigate the levels of endotoxins in teeth with apical periodontitis and determine the inflammatory mediators.	IL-1β and TNF-α were reduced when the levels of endotoxins decreased. These were minimised because of chemo-mechanical debridement of the root canal and, consequently, less activation of the proinflammatory cells such as macrophages.
Periapical fluid RANKL and IL-8 are differentially regulated in pulpitis and apical periodontitis. (Rechenberg et al., 2014) [19]	IL-8	Research the levels of RANKL, OPG, and IL-8 in periapical tissue fluid of human teeth diagnosed with irreversible pulpitis and apical periodontitis.	Results suggest that periapical bone resorption, determined by RANKL, occurs before inflammatory cell recruitment signalling, determined by IL-8.
Pulp Inflammation Diagnosis from Clinical to Inflammatory Mediators: A Systematic Review. (Zanini et al., 2017) [34]		Review inflammatory mediator expression in the context of clinical diagnosis to evaluate pulp inflammation severity.	Clinical irreversible pulpitis is related to specific levels of inflammatory mediator expression. The difference between reversible and irreversible is both quantitative and qualitative.
Inflammatory cytokines in normal and irreversibly inflamed pulps: A systematic review. (2017, Hirsch et al.) [35]	IL-6, IL-8, IL-2, TNF- α.	Review literature regarding the inflammatory process and pulpitis	inconsistencies between studies exist and therefore, it is difficult to select just one specific cytokine suitable for testing. Some cytokines mentioned: IL-6, IL-8, IL-2, TNF-α.

**Table 2 diagnostics-14-01099-t002:** Comparison between pulp’s state and types of cytokines.

Healthy Pulp/Reversible Pulpitis	Irreversible Pulpitis (Increased Levels)	Pulpal Necrosis/Apical Periodontitis (Increased Levels)
IL-4	IL-1 β	IL-1
IL-10	IL-2 *	IL-6
Low levels of IL-2 *	IL-6	IL-8
Low levels of IL-8	IL-8	IL-17
Low levels of TNF-α	TNF-α	TNF-α
MMP-3	INF-y	TFG-β
	MMP-9	RANK-L/OPG system
		MMP-9

* IL-2 shows inconsistency and authors believe it should not be included as a possible biomarker.

## Data Availability

Data will be available by corresponding authors.

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
