# Peer review of "Understanding the Dynamics of Inflammatory Cytokines in Endodontic Diagnosis: A Systematic Review"

_diagnostics, 2024, doi:10.3390/diagnostics14111099_

Round 1

Reviewer 1 Report

Comments and Suggestions for Authors

Hello, dear authors!

Thank you for the opportunity to review your review. In general, the impression is favorable, there are no comments on the design.

It will be useful to formulate a null hypothesis and control points for the study. The review methodology was compiled correctly, there are no questions about the time range.

Reviewer 2 Report

Comments and Suggestions for Authors

Dear authors,

this is a very interesting review that could find some applications in daily practice, if further developed. The review is well conduced but there are some comments to address, to improve the quality and the scientific presentation of your work. 

Introduction and materials and methods are ok. 

Results must be better exposed; it is not enough to present only a single table. Please explain better each study and why do you have considered it in your review. 

Discussion section must be improved and reorganized. There is a lack of a specific comparison of your results with the current literature. Is quite general and doesn’t focus on the main aim of your study. 

Moreover, I suggest to provide a table or a fig with the cytokines involved in the different phases of pulpal inflammation, in order to have a simple visualization of the cytokines pattern changing within the pulpal tissue from reversible pulpitis to pulpal necrosis.  

Thank you.

Comments on the Quality of English Language

minor editing is required
